# Gum Acacia-Crosslinked-Poly(Acrylamide) Hydrogel Supported C_3_N_4_/BiOI Heterostructure for Remediation of Noxious Crystal Violet Dye

**DOI:** 10.3390/ma15072549

**Published:** 2022-03-30

**Authors:** Gaurav Sharma, Amit Kumar, Mu. Naushad, Pooja Dhiman, Bharti Thakur, Alberto García-Peñas, Florian J. Stadler

**Affiliations:** 1Shenzhen Key Laboratory of Polymer Science and Technology, Guangdong Research Center for Interfacial Engineering of Functional Materials, Nanshan District Key Laboratory for Biopolymers and Safety Evaluation, College of Materials Science and Engineering, Shenzhen University, Shenzhen 518055, China; mittuchem83@gmail.com (A.K.); fjstadler@szu.edu.cn (F.J.S.); 2International Research Centre of Nanotechnology for Himalayan Sustainability (IRCNHS), Shoolini University, Solan 173212, Himachal Pradesh, India; dhimanpooja85@gmail.com (P.D.); bt3456thakur@gmail.com (B.T.); 3School of Science and Technology, Glocal University, Saharanpur 247001, Uttar Pradesh, India; 4Instituto de Productos Naturales y Agrobiología, Consejo Superior de Investigaciones Científicas (IPNA-CSIC), Avda. Astrofísico Fco. Sánchez 3, 38206 La Laguna, Spain; 5Department of Chemistry, College of Science, King Saud University, Riyadh 11451, Saudi Arabia; munaushad@ksu.edu.sa; 6Departamento de Ciencia e Ingeniería de Materiales e Ingeniería Química (IAAB), Universidad Carlos III de Madrid, 28911 Leganés, Spain; albertga@ing.uc3m.es

**Keywords:** nanocomposite hydrogels, C_3_N_4_/BiOI heterostructure, adsorption, photocatalysis, crystal violet, Z-scheme

## Abstract

Herein, we report the designing of a C_3_N_4_/BiOI heterostructure that is supported on gum acacia-crosslinked-poly(acrylamide) hydrogel to fabricate a novel nanocomposite hydrogel. The potential application of the obtained nanocomposite hydrogel to remediate crystal violet dye (CVD) in an aqueous solution was explored. The structural and functional analysis of the nanocomposite hydrogel was performed by FTIR (Fourier transform infrared spectroscopy), X-ray diffraction (XRD), transmission electron microscopy (TEM), and scanning electron microscopy (SEM). The different reaction parameters, such as CVD concentration, nanocomposite hydrogel dosage, and working pH, were optimized. The C_3_N_4_/BiOI heterostructure of the nanocomposite hydrogel depicts Z-scheme as the potential photocatalytic mechanism for the photodegradation of CVD. The degradation of CVD was also specified in terms of COD and HR-MS analysis was carried to demonstrate the major degradation pathways.

## 1. Introduction

The presence of colored pollutants in the aquatic environment deteriorates the quality of water and negatively impacts flora and fauna. The potential sources of these effluents are from various industries such as printing, textile, food, cosmetics, and leather, etc. [1,2]. Crystal violet dye (CVD) is a major colorant that is used in the textile industries, and its release into natural water bodies without treatment leads to water pollution. The filthy water with CVD causes different health risks, showing harmful effects on the liver and kidney, causing skin irritation, and even some studies revealing its carcinogenic activity too [3,4,5,6]. Thus, it necessitates removing such dyes from wastewater. The remediation of polluted water includes practices such as chemical oxidation, membrane filtration, ion exchange, photocatalysis, coagulation, adsorption, and flocculation, etc. [7,8,9,10,11,12,13]. Even bioremediation of noxious pollutant use different microorganisms such as *Agrobacterium radiobacter, Saccharomyces cerevisiae*, and *Enterobacter* etc. [14,15,16]. All these practices offer a good deal for handling polluted water, but inherent problems exist in their implementations that are related to cost-effectiveness and operation. The synergism of these practices can open a new avenue in the field of polluted water remediation [17,18]. Recently, adsorption and photocatalysis have been presented by researchers to develop a new class of adsorptional-photocatalytic materials. Such materials demonstrate the synergism between practices leading to enhanced results for the removal of contaminants from wastewater [19]. Thus, to propagate the adsorptional-photocatalytic technique, research has been extensively carried out to develop and design such materials [20,21].

Recently, bismuth oxyhalide (BiOX; X = F, Cl, Br, I) emerged as a prominent material in the field of photocatalysis owing to its excellent photoactivity under the visible spectrum of solar light. BiOI possesses layered structures of (Bi_2_O_2_)^2+^, which merges in a double slab of iodine atoms with a narrow band gap of 1.63~2.1 eV [21,22,23]. Similarly, C_3_N_4_ demonstrated potential photocatalytic activity and is well recognized as a metal-free photocatalyst.

A range of materials were used as support for adsorbents, for example, activated charcoal, biochar, and hydrochar, etc. These materials provide additional surface area that is enriched with various functionalities. Hence, these materials not only provide support for catalysts but also increase the efficiency by adsorbing pollutants. Hydrogels are known as three-dimensional polymeric crosslinked biogenic materials that are being explored in diverse fields such as medical, cosmetics, and most recently for environmental detoxification [24,25,26,27]. These possess abundant hydrophilic functional groups on their surface, such as -OH and -COOH. Due to these hydrophilic groups, water easily diffuses into the hydrogel due to capillary action and differences in osmotic pressure [28,29].

Various types of natural polymers such as starch, pectin, cellulose, alginate, gelatin, chitosan, gum acacia, and guar gum, and synthetic polymers such as polyvinyl pyrrolidone, and polyvinyl alcohol have been explored for the synthesis of hydrogels [30,31,32,33,34]. Gum acacia can be extracted from the stem and branches of the Acacia Senegal tree. Among all these natural polymers, gum acacia is suitable for hydrogel fabrication, and it improves the water retention property of hydrogels. It shows quite promising properties for its utilization in the preparation of hydrogel, such as biocompatibility, antioxidant, antibacterial, and anti-inflammatory properties. It is highly soluble in water with a slightly viscous texture [35]. Chemically, gum acacia is composed of 4-O-methyl glucuronic acid units that are joined with β-glycosidic linkage. It contains D-galactose, D-glucuronic acid, L-arabinose, and L-rhamnose at an appropriate molar ratio [36]. It also contains 3% proteins (serine, alanine, isoleucine, threonine, valine, tyrosine, methionine, cysteine, and hydroxyproline) [37,38].

In this investigation, a C_3_N_4_/BiOI heterostructure that was supported on GA-cl-poly(acrylamide) hydrogel was fabricated to remove CVD from water based on the above advantages. The fabricated nanocomposite hydrogel displays the properties of both parts, i.e., hydrogel and photocatalyst. The above hydrogel’s backbone is composed of gum acacia and poly(acrylamide), which trigger the adsorption process on the surface. Also, the C_3_N_4_/BiOI photocatalyst gets entangled into the polymeric matrix, inhibiting its agglomeration. Overall, it leads to enhanced efficiency of the nanocomposite hydrogel as an adsorptional-photocatalyst, which ultimately results in better removal of noxious CVD.

## 2. Materials and Methods

### 2.1. Materials

Gum acacia, ammonium persulphate, N,N-bismethyleneacrylamide, ethylene glycol (Loba Chemie, Mumbai, India), acrylamide, urea, potassium iodide, bismuth nitrate, crystal violet dye (Chemical Drug House, India), and double-distilled water were utilised. All the chemicals that were used were of analytical grade.

### 2.2. Synthesis of Nanocomposite Hydrogel

#### 2.2.1. Synthesis of BiOI

In a typical experiment for the synthesis of BiOI, 10 mL of an aqueous solution consisting of 20 mg KI was added dropwise to 10 mL ethanol containing 50 mg Bi(NO_3_)_3_. The mixture was then mixed properly using vigorous stirring until the color changes from yellow to red. The resulting precipitates were collected and washed using ethanol and distilled water several times and then dried in air at 50 °C for 5 h in a hot air oven [39].

#### 2.2.2. Synthesis of C_3_N_4_

C_3_N_4_ was prepared by the thermal annealing of urea. For this, 100 g of urea was loaded to a silica crucible and calcined at 450 °C in a muffle furnace at a heating rate of 5 °C/min. When it reached the peak temperature, heating was continued for another 3 h for complete de-ammoniation. Finally, then it was kept to cool-down to room temperature. The obtained fluffy yellow powder was then washed with water/ethanol solution and dried overnight at 50 °C. The C_3_N_4_ powder was stored in an airtight container for further use [40].

#### 2.2.3. Synthesis of GA-cl-poly(acrylamide)@C_3_N_4_/BiOI Nanocomposite Hydrogel

For the synthesis of nanocomposite hydrogel, firstly, two solutions were prepared. Solution A was obtained by mixing C_3_N_4_ and BiOI in 10 mL distilled water at a molar ratio of 1:1. In solution B, 0.500 g of gum acacia was dissolved in 50 mL distilled water using stirring and 20 mL (0.2M) acrylamide was added to it. Solutions A and B were then mixed. To this mixture, 8% N,N-bismethyleneacrylamide and 5% ammonium persulphate were added with constant stirring at room temperature. The blended mixture was then constantly mixed for 2 h and then treated in a microwave oven at 40 W for 3 min. The obtained gel was then washed with distilled water several times and dried overnight in a hot air oven at 40 °C.

### 2.3. Swelling Studies

The swelling studies of the obtained nanocomposite hydrogels were performed to understand the effect of varying ratios of C_3_N_4_ and BiOI (1:1, 1:0.5, 0.5:1, 1:2, and 2:1) in its structure. A total of 0.100 g of the oven-dried synthesized samples were then immersed in distilled water for 24 h. The samples were then dried using filter paper to remove the extra water. The final weight of the samples was then noted, and % swelling was calculated using the following Equation (1) [41]:(1)$$\%\;\mathrm{swelling}=\frac{W_{t}-W_{d}}{W_{d}}\times 100$$
where *W_d_* is the initial weight of the dried hydrogel, and *Wt* is the weight of the swollen hydrogel.

### 2.4. Characterization

The FTIR spectrum was obtained in the range of 400–4000 cm^−1^ to confirm the presence of various functional groups (Agilent Technologies, L1600312 TWOLITA/ZnSe). X-ray diffraction (XRD) spectra of the samples were performed at room temperature, using a Bruker D8 advance X-ray diffractometer employing CuKα radiation (2θ scan rate of 2° min^−1^ in the 2θ range of 5–70°). The UV-Vis spectra were obtained (Perkin Elmer, Lambda 650 s) using BaSO_4_ as a reference. The topographic images were obtained by transmission electron microscope (TEM) (FP 5022/22-Tecnai G2 20 S-TWIN, Hillsboro, OR, USA), and the surface image was obtained by a scanning electron microscope (SEM) (JEOL JEM-2100F). Thermal analysis was performed by the thermogravimetric analyzer (Q500 TA Instruments, USA). The nanocomposite was heated at 10°C/min in a nitrogen atmosphere from 10 to 1000 °C.

The zero point charge (pHpzc) was determined by the pH drift method [42]. The optical studies were carried out using Tauc plots and the absorbance was analyzed in the UV-Vis range. A suspension was prepared by adding 0.005 g of the synthesized nanocomposite hydrogel in 10 mL ethanol, followed by sonication for 1 h. Afterwards, the UV-Vis spectrum was recorded in the range of 200–800 nm using a double beam spectrophotometer. The band gap was calculated by using the Tauc relation (Equation (2)) [43]:αhν= B (hν-Eg)_n_(2)
where α is the absorption coefficient (2.303 A/l), Eg is the optical band gap, B is the band tailing parameter, hν is the photon energy, and n is determined by the type of optical transition through PL analysis.

### 2.5. Remediation of CVD Using GA-cl-poly(acrylamide)@C_3_N_4_/BiOI Nanocomposite Hydrogel

Remediation of CVD was carried out to examine the adsorption-photocatalytic activity of the nanocomposite hydrogel. For this, 0.020 g of the nanocomposite hydrogel was suspended in 30 ppm CVD solution. The remediation experiments were conducted under three different protocols. In the first protocol, the mixture was kept in the dark for 1 h to attain equilibrium in the adsorption-desorption process. Then, the mixture was kept under sunlight (i.e., traditional photocatalysis). In the second protocol, the mixture was directly kept under sunlight (adsorptional-photocatalysis), and in the third protocol, the removal of CVD was studied under complete dark conditions (adsorption). Under each protocol, an aliquot of 2 mL was taken out after a fixed interval of time and analyzed by a UV-Vis spectrophotometer. The decrease in the intensity of CVD (lambda max at 591 nm) was recorded as a function of time.

The % remediation of CVD was calculated using the following Equation (3) [44]:(3)$$\%\;removal=\frac{C_{0}-C_{t}}{C_{0}}\times 100$$
The kinetics for the removal of CVD was illustrated by the pseudo-first-order kinetics. The rate constant (*k*) for the dye was calculated using the following Equation (4).
(4)k=2.303×slope
where the slope was obtained from the graph (ln C_0_/C vs t).

Chemical oxygen demand (COD) estimation is an important parameter for evaluating water quality. The COD value for the treated water (after the degradation experiment) was determined by the mercury sulphate method. The COD was calculated using the following Equation (5).
(5)$$COD=\frac{\left(A-B\times N\times 8\times 1000\right)}{Volumeofsample\;taken}$$
where A and B are the volume of Fe(NH_4_)_2_(SO_4_)_2_ for the blank and the sample, respectively, N is the normality of titrant. HR-MS analysis was carried out to ensure the complete mineralization of CVD to elucidate the possible mineralization path.

## 3. Results and Discussion

### 3.1. Synthesis of GA-cl-poly(acrylamide)@C_3_N_4_/BiOI Nanocomposite Hydrogel

Various samples of GA-cl-poly(acrylamide)@ C_3_N_4_/BiOI were synthesized by the microwave method. N,N-methylenebisacrylamide and ammonium persulphate were used as cross-linker and initiator, respectively. The microwave treatment assisted in providing the required temperature conditions for the generation of reactive radicals. These radicals further aided in the crosslinking of the units that were present, and C_3_N_4_ & BiOI units also attached by H-bonding and π-π interactions. The schematic exhibiting the synthesis of GA-cl-poly(acrylamide)@C_3_N_4_/BiOI nanocomposite hydrogel is shown in Figure 1.

The GA-cl-poly(acrylamide)@ C_3_N_4_/BiOI nanocomposite hydrogel attained the maximum swelling capacity of 180% at a 1:0.5 ratio of C_3_N_4_:BiOI, as shown in Figure 1a. The other ratios might increase the extent of interactions with the polymeric units which decreased the penetration of water molecules into their system and ultimately decreased the swelling percent.

### 3.2. Characterization Results

Figure 1b shows the FT-IR spectrum of BiOI (Figure 1b, part (i)) and GA-cl-poly(acrylamide)@C_3_N_4_/BiOI (Figure 1b, part (ii)) nanocomposite hydrogel. The peaks for the nanocomposite hydrogel at 3079 cm^−1^ and 3208 cm^−1^ indicate -OH stretching [45]. The intense peak that appeared at 1660 cm^−1^ is due to the presence of a C=O group [46]. Amide II & III groups in (Figure 1b, part (ii)) are confirmed from the peaks that appeared at 1413 cm^−1^ and 1455 cm^−1^ [47]. The peak for the C-OH group was observed at 1045 cm^−1^. For C_3_N_4_, the peak at 2831 cm^−1^ corresponded to the stretching vibration of the aromatic C-N group. Also, the presence of heptazine rings was ascertained from the appearance of peaks at 781 cm^−1^ and 611 cm^−1^ [48,49]. Bi=O=Bi vibration in the nanocomposite hydrogel was observed at 848 cm^−1^ as compared to 883 cm^−1^ in the case of pristine BiOI in (Figure 1b, part (i)). Similar changes occurred for the Bi-O stretching vibration. In the case of pristine BiOI, the peak occurred at 472 cm^−1^ and shifted to 530 cm^−1^ in the case of the nanocomposite hydrogel, suggesting its successful fabrication [50,51].

Figure 1c confirms the crystal structure of the synthesized nanocomposite hydrogel. The XRD peaks at 2θ = 29.6°, 31.7°, 39.4.1°, 45.8°, 51.5°, 55.1°, 66.2°, 55.1°, and 75.3°are indexed to (012), (110), (004), (014), (114), (122), and (220) of BiOI. These highly intense and sharp peaks confirm the presence of crystalline BiOI and tetragonal structure of BiOI (JCPDS 10-0445), while the diffraction peak at 2θ = 27.5° is indexed to (002) of carbon nitride [52,53]. The appearance of peaks at 2θ = 20.7°, 26.4°, and 37.1° demonstrate the presence of gum acacia and polyacrylamide. Also, it has been noted that the crystallite size of the synthesized sample is 27.3 nm.

Figure 1d shows the TGA and DTG plots of GA-cl-poly(acrylamide)@C_3_N_4_/BiOI nanocomposite hydrogel. Initially, the weight loss was observed between 20 and 170 °C. This may be attributed to the loss of additional water in the nanocomposite hydrogel. The weight loss from point (ii) to (iii) maybe due to the disruption of the polysaccharide backbone between (170–250 °C). The weight loss in the range of 230–490 °C is assigned to the elimination of the amine group. At the end of the process, CO_2_ was liberated, and the weight loss was estimated at 550 °C. The DTG curve shows the maximum weight loss at 505 °C.

The SEM micrographs depicted the surface morphology of the nanocomposite hydrogel. Figure 2a shows the smooth surface of GA-cl-poly(acrylamide)@C_3_N_4_ nanocomposite hydrogel. The C_3_N_4_ layered sheets were observed. In Figure 2b,c, the circular-shape of the material confirms the presence of BiOI photocatalyst onto the surface of nanocomposite hydrogel [54]. Figure 2d–f indicates the TEM images of GA-cl-poly(acrylamide)@ C_3_N_4_/BiOI nanocomposite hydrogel at different magnifications; the C_3_N_4_ layered sheets networks are clearly visible with some dark regions reveling presence of agglomerated BiOI particles which are in good agreement with the SEM data. Due to presence of hydrogel and agglomeration in composite it was difficult to mark individual BiOI particles, but the average particle size of BiOI particles was observed between 20–50 nm.

### 3.3. Band Gap and Point of Zero Charges (pHpzc) Study

Figure 3a depicts the Tauc plot of C_3_N_4_ and BiOI. The band gap of BiOI and C_3_N_4_ are 1.89 and 2.70 eV, respectively. The band gaps of the nanocomposite hydrogel components emphasize their visible light activeness. Figure 3b shows that the pHpzc of the nanocomposite hydrogel is 6.6. This indicates that when pH < pzc, the surface is positively charged, while pH > pzc, it is negatively charged. The study of pHpzc is crucial as removing any pollutant molecule and is directly linked to the surface properties of the adsorbent or catalyst. Due to surface charge, the molecules exhibited electrostatic interactions and hence the pollutant and catalysts play an important role.

### 3.4. Parameters for the Remediation of Crystal Violet Dye

Crystal violet dye is also known as triarylmethane dye and can be used to classify bacteria in the Gram’s method. Figure 4 shows different optimized parameters for the removal of CVD under sunlight. Figure 4a shows the effect of pollutant concentration on removing CVD in the range of 10–50 ppm. The result shows that the % of removal gradually increased from 51 to 82% from 10 to 30 ppm. But a decrease in the degradation of CV was observed (58%) above 30 ppm. This might be because the pollutant molecule might occupy all the active sites at 30 ppm, and very little remains free to accommodate more CV molecules. Figure 4b shows the effect of the nanocomposite hydrogel dosage on the % removal in the range of 10–50 mg/30 ppm 100 mL of CVD solution. The maximum removal was found to be 83% at 20 mg of GA-cl-poly(acrylamide)@ C_3_N_4_/BiOI nanocomposite hydrogel dosage. With an increase in the dosage above 20 mg, the removal starts to decrease. This could be due to that the path of light being blocked by the excessive amount of nanocomposite hydrogel, which reduces the radiation’s penetration into the solution. Figure 4c exhibits the effect of pH on the removal, and it is evident that the maximum removal occurs in the basic pH. The effect was studied in the pH range of 2–14. A total of 80% removal was achieved in 140 min at pH 9. This can be explained based on the PZC. As nanocomposite hydrogel possesses a PZC of 6.6, when the pH < PZC the surface of the nanocomposite is positively charged, and when pH > PZC the surface becomes negatively charged. Thus, a very strong electrostatic attraction occurs at a higher pH between the nanocomposite and the negative surface of the CV dye, which is cationic. However, after pH 9, the percent removal starts to decline. This is because at a higher pH, the concentration of OH- increased, which may compete for adsorption with the pollutant molecules at the nanocomposite hydrogel’s surface. Figure 4d shows the effect of H_2_O_2_ on degradation in the range of 2 to 1.4 mM. H_2_O_2_ gives ^●^OH radicals by exposing to sunlight. These ^●^OH radicals are the most powerful oxidizing species which can destroy any organic molecules. The maximum removal of 71% was observed within 80 min of exposure to sunlight at 0.8 mM, while a decline was observed above this.

### 3.5. Adsorption-Photocatalysis of CVD

The potential of GA-cl-poly(acrylamide)@C_3_N_4_/BiOI nanocomposite hydrogel was investigated to remediate CVD under natural sunlight. Figure 5a shows the % removal under three different conditions; photolysis, adsorption in dark, and adsorptional-photocatalysis. In the case of photolysis, a maximum of 6% removal was observed even after 180 min, indicating the stable nature of CVD. Under dark conditions, the removal was completely dominated by the adsorption process. The hydrogel part may offer the required reactive sites and surface area that resulted in 46% removal of the CVD molecules. A maximum removal rate of 88% was obtained in adsorptional-photocatalysis, where the adsorption and photocatalysis processes occurred simultaneously. The hydrogel part offered various interactions to the CVD molecules, and the photo-generated reactive radicals assisted in the mineralization of the adsorbed CVD molecules.

Figure 5b exhibits the % removal of CVD in the case of combination of adsorption and photocatalyst (traditional photocatalysis). The reaction solution was first placed in the dark for attaining the adsorption-desorption equilibrium for 60 min and then kept in the light for photocatalysis. As presented, a maximum adsorption rate of 44% was observed within 60 min that could be linked to various functionalities that are present on the nanocomposite hydrogel’s surface. However, when placed in the light, the maximum removal was 74%. Here, as compared to the adsorptional-photocatalysis, a lower rate was obtained. This could be linked to the blockage of some active sites through adsorption that might have reduced the production of reactive radicals, thereby decreasing the removal rate. Figure 5c shows the pseudo-first-order kinetic plots under three conditions (as given in Section 2.5). The apparent rate constant (k_1_) for the adsorptional-photocatalysis was 0.0299 min^−1^, while for other conditions, i.e., adsorption + photocatalysis and dark, it was 0.0253 and 0.004 min^−1^, respectively as shown in Table 1.

The study involving scavengers was carried out to identify the main reactive species that are involved in CVD degradation by the nanocomposite hydrogel. For this experiment, different radical scavengers such as EDTA, potassium dichromate, 1,4-benzoquinone, and isopropyl alcohol were employed for scavenging the activity of h^+^, e^−^, ^*^O_2_^−^, and ^*^OH, respectively [44]. Figure 5d depicts the effect of the employed scavenger on the photodegradation of CVD in the presence of the nanocomposite hydrogel. The results illustrate that the addition of IPA reduced the degradation rate to a greater extent, indicating that the ^●^OH radicals are the major participating species in the degradation of CVD. The second species in which the rate reduced remarkably was the 1,4 benzoquinone, indicating the contribution of ^●^O_2_^−^ radicals. Thus, the majority of the active radical species that are responsible for photodegradation of CVD are ^●^OH and ^●^O_2_^−^. The contribution of different species in the CVD degradation can be indicated as follows: ^●^OH > ^●^O_2_^−^ > h^+^ > e^−^.

### 3.6. Mechanism of Photodegradation

The nanocomposite hydrogel implicates its excellent potential for photocatalytic activities. The mechanism of CVD degradation can be described by looking into the constituent photocatalysts, i.e., C_3_N_4_ and BiOI. These are known for their excellent band gaps, which are visible light active. The in situ synthesis of GA-cl-poly(acrylamide)@C_3_N_4_/BiOI nanocomposite hydrogel exhibited a favorable band structure that was formed between both the photocatalysts. Considering the charge transfer mechanism, two probable mechanisms are possible; (a) traditional heterojunction and (b) Z-scheme. When the nanocomposite hydrogel was subjected to solar radiation, it absorbs the energy, and the excitation of electrons occurs by forming holes. Focusing on the charge transfer, in the case of traditional heterojunction, the electrons move from the CB of C_3_N_4_ to BiOI and holes transfer follow the reverse path. As a result, the holes accumulate on the VB of C_3_N_4,_ possessing a potential of +1.53 eV. However, for the successful formation of the ^●^OH radicals, a minimum potential of +1.99 eV is required, suggesting that some other pathway must be followed that assists in forming the ^●^OH radicals (major reactive species as indicated by the scavenging study). However, in the Z-scheme, charge transfer occurs from the CB of BiOI to VB of C_3_N_4_. As a result, the higher potential valence and conduction bands are free to form the required reactive radicals. These formed radicals then assist in the mineralization of CVD dye. The complete charge transfer using the two different mechanisms are presented in Figure 6.

Furthermore, the degradation pathway of CVD is studied by HR-MS. Figure 7a,b shows the HR-MS of CVD before and after the degradation. The main fragments were detected at m/z: 341.1392, 327.1247, 281.2154, 255.2154, 224.9669, 194.9048, 157.1046, 127.9950, and 96.94. Based on the HR-MS analysis, a tentative mechanism is proposed, as shown in Figure 2. The ^●^OH radicals (the main reactive species as confirmed by the scavenger study) majorly attack CVD. The attack involves the formation of various intermediates through a series of reaction steps. Finally, the ring-opening and parent molecules degrade into low molecular weight aliphatic compounds and, ultimately, mineralize into water and CO_2_.

### 3.7. COD Analysis and Evaluation of Reusability

Figure 8a shows the % removal in terms of COD values. The COD is reduced from 100 to 21% after 180 min of degradation. This confirms the mineralization of CVD into smaller compounds. A reusability study is important for any photocatalysts to ensure their real-time application for wastewater treatment. To examine the reusability, the nanocomposite hydrogel was separated from the aqueous solution of the pollutant by centrifugation and washed with ethanol/water mixture thoroughly and dried for 3 h at 50 °C. Afterwards, under similar conditions, the same photodegradation experiment was performed for four consecutive cycles. The degradation percent fell from 88 to 68% from the first to fifth cycle, as depicted in Figure 8b. The decline may be due to the blockage of active sites at the surface of nanocomposite through various degraded intermediates.

## 4. Conclusions

In this investigation, biopolymer (gum acacia)-based nanocomposite hydrogel with BiOI and C_3_N_4_ photocatalyst was used to remove crystal violet dye (CVD) from water. The maximum swelling capacity of 180% for the nanocomposite hydrogel was observed. The synthesized nanocomposite hydrogel was found to be visible light active for the photocatalytic activities with the band gap energy of 1.89 and 2.7 eV. The remediation study confirms that the nanocomposite hydrogel is capable for the removal of 88% CVD within 180 min, while in the dark, 46% removal was observed. The hydroxyl radicals are identified as the main reactive species by the radical scavenging experiments. The COD and HR-MS analysis implicate that the pollutant molecules are mineralized into small molecular weight aliphatic compounds. Overall, this study might attract research in exploring biopolymer-based nanocomposite hydrogel for adsorption-photocatalysis based on various environmental detoxification applications.

## Data Availability

Not applicable.

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
