# Peer review of "Gum Acacia-Crosslinked-Poly(Acrylamide) Hydrogel Supported C3N4/BiOI Heterostructure for Remediation of Noxious Crystal Violet Dye"

_materials, 2022, doi:10.3390/ma15072549_

Round 1

Reviewer 1 Report

Thank you for making the suggested changes. I think, the manuscript is now ready to be accepted.

Author Response

Thank you so much for accepting our work.

Reviewer 2 Report

Authors have replied the reviewers’ comments and have made some modifications in the manuscript, but the reviewers is still not satisfied. There is still poor scientific approach, why a single concentration of the individual components have been used? Authors need to conduct more experiments for optimization of the concentrations of all these components and they should provide experimental evidences for choosing a particular concentration. In this context, I must reject this manuscript for publication in Materials.

Author Response

Thank you for your valuable comment and rejection  as we are working on novel adsorptional-photocatalysts we are not focusing on individual application as adsorption (i.e., only for hydrogel and  C3Npart) or photocatalysis (i.e, for BiOI and C3Npart) so making comparison for individual components will create lot of chaos and mess so we did not included it. Instead of that we have given comparison of two different processes for same material that is adsorption on dark,   adsorptional-photocatalysts (i.e., first adsorption in dark followed by photocatalysis in light and lastly photocatalysis (i.e., performing direct photocatalysis in light). On the other hand you did not follow that we have opatized the ratio of photocatalyst and hydrogel to balance the two applications. I hope you can understand our concept presentation. Surely we will work on valuable suggestions in our upcoming works where we will compare individual components with different processes. And as of covid restriction we have limited access to lab and resources at present. Thank you. 

Reviewer 3 Report

In my opinion, the submitted manuscript is suitable for publication after minor corrections.
- in the introduction, you can add one of the possibilities of using microorganisms as components of sorbents, which can be used as a sorbent for removing dyes. Here is an example of the use of a biocomposite as a sorbent doi.org/10.3390/ma14237482 There are microorganisms capable of breaking down crystal violet.
- there are some editing errors in the manuscript, so I suggest checking the text again (Line 130 - Wd "d" should be in the index, 137 - m-1 "-1" index, 138 - BaSO4 "4" index, 171 - C0 "0" index, in equation 5 replace the multiplication sign with "*" for the multiplication sign)
- figure 1b - in the description of the X axis it is "Wavelength" and it should be "Wavenumber", no description of which spectrum is from what (this information is not available in the text or in the drawing description)
- figure 1d - the description of the X axis is Time given in degrees Celsius and it should be in minutes, additionally, after "min" there is no dot "min." . Additionally, once a unit is written in parentheses and once without.
- Figure 6 is of poor quality. It's a bit fuzzy.
- Figure 7 is of tragic quality. It needs to be corrected because you can't see anything on it.

Author Response

In my opinion, the submitted manuscript is suitable for publication after minor corrections.
- in the introduction, you can add one of the possibilities of using microorganisms as components of sorbents, which can be used as a sorbent for removing dyes. Here is an example of the use of a biocomposite as a sorbent doi.org/10.3390/ma14237482 There are microorganisms capable of breaking down crystal violet.

Answer: As per related comment we have added possibilities of using microorganisms as components of sorbents, which can be used as a sorbent for removing dyes suggested work has been included in text.

- there are some editing errors in the manuscript, so I suggest checking the text again (Line 130 - Wd "d" should be in the index, 137 - m-1 "-1" index, 138 - BaSO4 "4" index, 171 - C0 "0" index, in equation 5 replace the multiplication sign with "*" for the multiplication sign)

Answer: All the suggest errors has been corrected.

- figure 1b - in the description of the X axis it is "Wavelength" and it should be "Wavenumber", no description of which spectrum is from what (this information is not available in the text or in the drawing description)

Answer: The suggested correction has been made.

- figure 1d - the description of the X axis is Time given in degrees Celsius and it should be in minutes, additionally, after "min" there is no dot "min." . Additionally, once a unit is written in parentheses and once without.

Answer: Thank you for suggested correction we have made changes.

- Figure 6 is of poor quality. It's a bit fuzzy.

Answer: The resolution of figure 6 has been improved as suggested.

- Figure 7 is of tragic quality. It needs to be corrected because you can't see anything on it.

Answer: Dear Reviewer figure 7 is instrument generated still be have tried to improve its resolution

All changes has been marked in red.

Round 2

Reviewer 2 Report

Authors have responded the issues raised by the reviewers but they failed to revise the manuscript according reviewers' comments. The reviewer questions are valid that by changing the contents of  C3N4 and BiOI  may effect the performances (adsorption and photocatalysis) of the composite hydrogels. Therefore, these issues must be addressed before publishing this work.